# OpenReview forum: "Keep CALM and Avoid Harmful Content: Concept Alignment and Latent Manipulation Towards Safer Answers"
_ICLR.cc/2026/Conference — ICLR 2026 Conference Withdrawn Submission_

### Official Review · Reviewer_cXuY · 2025-10-24

**Soundness:** 2
**Presentation:** 2
**Contribution:** 2
**Rating:** 2
**Confidence:** 2

**Summary:**

The authors propose CALM, a technique that filters harmful LLM responses at test-time.
In addition to improved safety, CALM's decisions are interpretable by mapping model activations to human-readable semantic concepts.

**Strengths:**

* The use of singular value decomposition in the embedding space to reject unsafe concepts is interesting and (to my knowledge) novel.

**Weaknesses:**

My main concern about this work is lack of empirical results:
* Specifically, the paper only compares CALM with ProFS which is conceptually similar.
* Additionally, the paper does not report how CALM impacts performance on benign datasets such as MMLU.

A relatively similar approach has been proposed in prior work:

Chen, Xin, Yarden As, and Andreas Krause. "Learning Safety Constraints for Large Language Models." In Forty-second International Conference on Machine Learning.

**Questions:**

* How does CALM compare with other state of the art methods in terms of safety and performance on benign tasks?

---

### Official Review · Reviewer_uJGV · 2025-10-31

**Soundness:** 3
**Presentation:** 1
**Contribution:** 2
**Rating:** 4
**Confidence:** 4

**Summary:**

This work studies the harmful content avoidance in LLMs. The authors propose CALM -- an inference-based method operating on the last-layer activations.  CALM uses a whitening strategy to align harmful/harmless directions with canonical axes at the latent space. Then, CALM projects out selected harmful components before decoding to avoid the harmful content generation. The authors evaluate on multiple datasets and LLMs to demonstrate the effectiveness.

**Strengths:**

1. The studied problem is timely and critical to ensure the safety of the LLMs.

2. The proposed method is practical and does not require additional training.

3. The experiments demonstrate some empirical improvements.

**Weaknesses:**

1. Technical novelty is limited, as the main method is primarily a simple adaptation of Uppaal et al. (2024), and Chen et al. (2020).


2. The presentation is a bit too wordy that hard to follow. The notation system is also complicated.


3. No comparison with existing methods, including both training required methods mentioned in the related work, as well as other inference time methods such as:
- InferAligner: Inference-Time Alignment for Harmlessness through Cross-Model Guidance, EMNLP'24.
- Towards Inference-time Category-wise Safety Steering for Large Language Models, arXiv'24.

4. Important details are missing. For example, how to specify the number of C, and why could one consider each entry in C as a concept.

5. No experiments in regular benchmarks such as GSM8K etc.

**Questions:**

Please find the details in the Weakness section.

---

### Official Review · Reviewer_CvTb · 2025-11-01

**Soundness:** 2
**Presentation:** 2
**Contribution:** 2
**Rating:** 2
**Confidence:** 4

**Summary:**

The authors propose CALM (Concept Alignment and Latent Manipulation), an inference-time method for reducing harmful outputs in LLMs. CALM combines Concept Whitening (CW), originally developed for computer vision, with orthogonal projection to identify and suppress harmful concept directions in the model's latent space. The method modifies representations at the last decoder layer without requiring retraining. Experiments across LLaMA, Phi-3, and Gemma model families demonstrate reductions in harmful outputs compared to baseline models and the ProFS baseline, as measured by perplexity metrics, toxicity scores, and harmfulness classifications.

**Strengths:**

I think the paper tackles an important and timely problem in LLM safety (broadly, preventing compliance with harmful requests), and the core motivation of applying inference-time interventions without retraining seems practically valuable. The experimental scope is reasonably comprehensive, covering multiple OS model families (LLaMA, Phi-3, Gemma) and several variants (Instruct, Pretrain, Abliterated versions).The interpretability analysis in Section 4.4 /  Figure 3 provides useful insights into which concept axes activate for different types of harmful content. The visualization of how specific harmful concepts (e.g., "Bomb Making," "Identity Theft" - though, I'm confused why these concepts specifically are grouped on the same concept axis, if I'm not mistaken) activate at token-level granularity is genuinely interesting and could inform future safety mechanisms.The authors are appropriately transparent about limitations, acknowledging issues with evaluation metrics, model-specific tuning requirements, and the challenge of handling entangled concepts.

**Weaknesses:**

I'm also a little confused about some of the example prompts shown in the paper, which look less like jailbreaks and more like unmodified harmful statements. The sample "red" response in figure 1, for example, seems pretty harmless. This isn't overall major, since it does seem that the paper applies this method to actual jailbreaks / real harmful requests. I'm confused by this, because my understanding is that production models don't comply with harmful requests, and the example responses of harmful provocations in appendix K feel somewhat borderline/marginal. Further, looking at some examples closer, I feel confused about whether this technique is  really doing much - for example if we take Fig 8, the "bad" baseline behaviour, when the model is asked to give egs of ways to engage in illegal activities to make money quickly, is that the model provides (as far as I can tell) *entirely legal means of making money*. The authors suggest in their caption that the CALM model

The defense results in Table 10 test only 8 models across 4 datasets, and the paper acknowledges this is "preliminary." Given that defense mechanisms are central to the paper's value proposition, this feels underdeveloped. The prompt intervention comparison (Appendix G) is interesting but arrives too late; this analysis should be foregrounded since prompt-based defenses are the most practical baseline for practitioners. As far as I can tell, the claims of this paper should hinge on how successfully the defended models

**Questions:**

1. Can the authors provide more examples of harmful prompts & responses, and how your method actually changes the models' responses? I feel confused about what effect you're claiming and on what types of prompt?
2. Why are provocations / harmful statements a relevant domain to use this technique on, instead of jailbreaks? Did you run on common jailbreaks, and what were the results?

---

### Official Review · Reviewer_TNbo · 2025-11-01

**Soundness:** 3
**Presentation:** 2
**Contribution:** 2
**Rating:** 2
**Confidence:** 4

**Summary:**

# Summary

The paper presents a method that prevents a model from producing harmful responses. It build on prior work that projects a model's embeddings onto a safe subspace. The central change is that the authors apply Concept Whitening and Alignment before projecting onto a safe subspace. The approach aligns some of the dimensions with harmful concepts and then zeros out those dimensions in latent space before projecting back to the original embedding space. The approach identifies these directions through an SVD of example harmful and safe responses to example prompts and rotates the embeddings so that these directions align with the canonical axes. They show that filtering the harmful axes increases perplexity on harmful responses.

# Contributions

 * A novel inference-time intervention that blocks harmful responses
 * Evaluations that show improved performance of this method over a variant that only filters subspaces, but does not whiten or align the dimensions

**Strengths:**

Overall, this is an interesting method that combines several existing ideas and has promise for improved performance at blocking harmful outputs from language models. It takes advantage of an existing interpretability method, and the results indicate that it could have increased performance in comparison to direct filtering or steering. I like how it relies on an existing dataset of harmful and safe responses, but does unsupervised learning on that to extract more granular concepts to intervene on.

**Weaknesses:**

## Improvement over the baseline

The primary weakness of the work is that the method does not show substantial improvements over the baseline subspace filtering method in the generation experiments (Table 4): For toxicity, their approach improves 50% of the time, performs the same 25% of the time, and gets worse 25% of the time. For harmfulness, it improves 53% of the time and gets worse 47% of the time.

## Presentation

My primary concern is that the paper is not very self-contained and relies heavily on appendices. For example, a central contribution of the paper is the addition of whitening to a pipeline for removing subspaces from the model's representation. However, whitening as a method is described in an appendix, not the main body of the paper. Similarly, the implementation details for the methods are discussed through a reference to Chen et al. 2020, instead of summarized or described within this paper.

A related challenge with the presentation is that it does not make the similarities or differences from these two key prior works clear. In reviewing the paper, I had to spend substantial time looking at the prior papers in order to evaluate the differences. In order to meet publication standards, the paper should be reasonably self-contained so that a member of the community can reasonably understand the main ideas and relationship to prior work from the main body of the paper.

## Evaluation

 * The paper compares with a baseline method that does subspace filtering without whitening. This is a useful comparison, but, given that it is a part of the CALM pipeline, this is more of an ablation than a baseline. Stronger baselines that more directly evaluate performance are needed to establish the performance of this method over alternatives.
 * The paper describes a comparison with prompting, but defers the details to the appendix. The reason for this is that 'instruction-tuned models are strongly inclined to follow instructions' --- this is true, but they are still remarkably brittle as defense mechanisms. Deferring the data to an appendix seems misleading in this context.
 * The evaluations are against static benchmarks and show some promise. However, given the motivation to defend against jailbreaks, it is important to evaluate with an adaptive attack. Consider, e.g., using GCG or a similar method to attack the model.
 * The evaluations are somewhat indirect as they rely on preference for multiple-choice responses (or perplexity of those responses). At least one evaluation should look at free-form generation and evaluate something like attack success rate or the rate of harmful responses.

## Validation of Interpretability
 * The authors discuss the interpretability of the method substantially in the introduction and the abstract. However, the actual evidence for interpretability is quite weak (a single heatmap showing activations across their learned dimensions for 2 examples).

## Minor Comments

 * l.109: TYPO, constrain --> constrainS
 * Citation formatting incorrect, likely a case of \citet instead of \citep or something similar

**Questions:**

* Are there alternative baselines you could consider? E.g., it seems like representation engineering or alternative methods for latent steering of the model would be appropriate comparisons for your method. Alternatively, direct steering in the original space could be a good baseline for this work.
 * Can you clarify the similarities and differences from both ProFS and Chen et al. 2020? The descriptions of the method go back and forth between summarizing the approach and referencing prior work and it is hard to distinguish the differences.
 * Is there additional evidence you can provide for the interpretability of the method and the associated benefits?
 * How would you restructure the paper to make the method description clearer and self-contained?
 * Can you justify the projection onto the orthogonal complement of the normal answers with data? Normally, I would expect this to be done by centering the embeddings before analysis (in the original space).

---

### Note · Authors · 2025-12-02

**Comment:**

We would like to thank the reviewers for the time and effort dedicated to evaluating our submission, as well as for the thoughtful and constructive feedback. While preparing for the rebuttal phase, we discovered reproducibility issues in our results. These took several weeks to diagnose, with the most significant problem caused by inconsistencies in eigenvalue decomposition across different hardware and software environments, which substantially impacted the downstream performance of our method. As a result, we were unable to devote our full attention to preparing a complete and thorough rebuttal. Although we ran some new experiments to address the reviewers questions and concerns, we believe these efforts were not enough.

In light of this, and given the recent leak of author and reviewer identities and the subsequent handling of the situation, we believe it is best to withdraw our submission from this year’s ICLR. The absence of a discussion phase and the current reviewer scores further reinforce our decision to step back from consideration at this time.

We sincerely appreciate the reviewers’ efforts and thank the program committee for their work during this challenging review cycle.

**Withdrawal Confirmation:**

I have read and agree with the venue's withdrawal policy on behalf of myself and my co-authors.